# Early Adversity and Changes in Cortisol and Negative Affect in Response to Interpersonal Threats in the Laboratory

**DOI:** 10.3390/ijerph19105934

**Published:** 2022-05-13

**Authors:** Christopher J. Wendel, Jenny M. Cundiff, Matthew R. Cribbet

**Affiliations:** Department of Psychology, University of Alabama, Tuscaloosa, AL 35487, USA; cjwendel@crimson.ua.edu (C.J.W.); mrcribbet@ua.edu (M.R.C.)

**Keywords:** adverse childhood experiences, cortisol reactivity, negative affect reactivity

## Abstract

Adverse childhood experiences, such as abuse and neglect, are associated with poor health outcomes. This association may be partially explained by differences in stress physiology. Though most early adverse experiences occur within the context of interpersonal relationships, stress exposures manipulated in the laboratory rarely involve interpersonal interactions beyond the mere presence of others. This study examines whether adverse childhood experiences are associated with differences in affective and cortisol reactivity to two stressors which may more closely resemble the powerlessness and the lack of control characteristic of many adverse childhood experiences: a dominant (vs. submissive) interaction partner and lower (vs. higher) social status. We also manipulate social-evaluative threat as a test of whether these interpersonal stressors are more germane to stress reactivity associated with early adversity than the performance anxiety evoked by more traditional laboratory stressors, such as the Trier Social Stress Test. The results partially support the hypothesis that participants with greater early adversity may be more reactive to interpersonal stressors reminiscent of early adverse experience. Given the interpersonal nature of most adverse childhood experiences, conceptualizing and measuring associations with stress physiology in an interpersonal context may more closely capture the psychological and biological embedding of these early experiences.

## 1. Introduction

Adverse childhood experiences (ACEs), such as childhood abuse and neglect, have been associated with increased risk for poor mental and physical health [1,2,3], and alterations in affective and physiological stress responses are one mechanism thought to partially explain this association [4,5]. Consistently, early adversity has been associated with elevated affective responding and deficits in emotion regulation [6,7,8,9]. For example, Heleniak et al. [10] found that childhood maltreatment was associated with greater negative affect reactivity, emotional intensity, and persistence of negative emotions both concurrently and prospectively. Hence, affective responses to stressors may represent a direct path to poor mental health, as well as a potential influence on physical health [11]. Additionally, these state-level differences are associated with a greater propensity for chronic negative affect in adulthood, which contributes to chronic activation of stress response physiology [12]. Chronic activation of these systems may help explain the well-established associations between negative affect (e.g., anger) and cardiovascular disease [13].

In addition to negative affect, ACEs are thought to have a direct influence on physiological stress responses via neurodevelopmental alterations [14,15,16] in the hypothalamic pituitary adrenal (HPA axis) [17], that, over time, contribute to the development of disease. The primary biological marker of stress reactivity in the HPA axis is the hormone cortisol [18,19], and changes in cortisol in response to stress have been associated with later disease development [20]. Associations between ACEs and cortisol reactivity to stress have been observed in many studies. For example, a 2013 meta-analysis examining cortisol reactivity in the laboratory suggests that ACEs are associated with dysregulated cortisol reactivity across the lifespan [21]. However, the direction of dysregulated cortisol responses (i.e., sensitized versus blunted) is not uniform, and may be influenced by methodological factors, such as when and how ACEs are assessed, or the nature of the laboratory stress manipulations used to evoke reactivity [21]. For example, in a sample of 8 to 13-year-olds, recent (within the last 12 months) and lifetime exposure to violence was associated with greater cortisol reactivity to the Trier Social Stress Task [22]. Similarly, physical abuse has been associated with greater cortisol reactivity to the Cold Pressor Task in a sample of older adolescents (ages 9–16) [20,23]. However, young adults reporting more ACE exposure have also displayed blunted cortisol reactivity in response to psychosocial stress [24,25], a finding consistent with results of the above referenced meta-analysis [21], which also finds that ACEs tend to be associated with blunted cortisol reactivity to stress, especially when assessed in adult participants [21].

The most common methods of inducing cortisol reactivity involve exposure to physiological stressors (e.g., mild pain from cold water), environmental stressors (e.g., noise exposure), cognitive stressors (e.g., mental arithmetic), or paradigms such as the Trier Social Stress Test (TSST) which combine social and cognitive stressors in an evaluative performance task [26,27,28]. Previous studies have revealed the TSST is similarly effective in eliciting negative affective responses [29]. Although these tasks are generally effective in eliciting psychological and physiological stress responses [28], they may be less reliable in distinguishing patterns of negative affect and cortisol reactivity associated with early adversity. Most early adversities (abuse, neglect, violence in the home) are interpersonal in nature, involving an interaction between the participant and others. Interpersonal stressors that more closely resemble interpersonal features of these early experiences may show more reliable associations with ACEs compared to cognitive, physiological, or mixed paradigm protocols, because interpersonal features of early adverse experiences may be a conditioned stimulus associated with threat, and thus, elicit a conditioned response. For example, the threat of hostility and the attack characteristic of experiences of abuse are not likely to be elicited by the performance anxiety associated with a public speaking task such as the TSST.

Work in related areas, such as stress responding in individuals with PTSD, has shown that the specific nature of the stressor is important for capturing reliable differences in stress reactivity associated with past experiences. Standard laboratory protocols, such as the TSST, may elicit substantially different patterns of physiological reactivity compared to stress exposures specific to a conditioned stimulus. For example, individuals diagnosed with PTSD typically display either blunted or statistically equivalent physiological reactivity to standard laboratory stress manipulations compared to control participants [30]. However, individuals with PTSD display greater physiological reactivity when exposed to manipulations closely resembling their traumatic experiences (e.g., combat sounds), or are instructed to recall traumatic experiences (i.e., script-driven imagery) [31,32,33,34,35]. Further, prior investigations also suggest some generalization of these stress responses by showing that stimuli loosely resembling previous stressors (i.e., loud noises unrelated to combat) also elicit sensitized physiological responding in veterans with PTSD, but not control participants [36,37], and these differences in biological stress reactivity specific to learned fearful stimuli are also found when comparing monozygotic twins who differ in exposure to trauma [38].

These results may be explained by the Threat Salience Model [39,40], which posits that exposure to situations with a high threat of harm (e.g., abuse, exposure to violence during active combat or in the home) are associated with sensitized physiological reactivity to stressors with high threat salience, and lesser physiological reactivity (e.g., blunting) to unrelated or low-threat stressors. Given the mixed findings, novel stress protocols are needed to investigate whether early adversity is associated with stimuli-specific cortisol reactivity similar to those found in the PTSD literature.

Interpersonal stressors that specifically manipulate lack of control or devaluation may be particularly well-suited to uncover differences in stress reactivity associated with ACEs. For example, exposure to interpersonal dominance may resemble authoritarian and controlling relationships, which characterize many ACEs, such as family conflict and verbal and physical abuse [38,39,40,41]. Similarly, experiences of lower social rank may signal social subordination and devaluation reminiscent of many of these same early experiences.

The current study examines whether early adversity within the family (e.g., abuse, neglect, exposure to violence) is associated with changes in cortisol and negative affect in response to interpersonal stress in a sample of 160 young adults (median age = 21). We specifically manipulate constructs that challenge the participant’s agency (e.g., exposure to interpersonal control, relative status), similar to experiences of powerlessness, devaluation, and lack of control that are thought to characterize many forms of early adversity. We also manipulate social-evaluative threat, often thought to be the “active ingredient” in the TSST, which has been widely used to manipulate stress in past research [28]. The manipulation of social-evaluative threat serves as a test of whether, similar to findings in the PTSD literature, exposure to interpersonal stimuli specifically associated with early adversity may be more likely to reveal differences in negative affect or stress reactivity by early adversity than the commonly employed TSST. Because cortisol reactivity can be influenced by time of day and participants’ body mass, we control for these factors in our analyses. Further, both cortisol reactivity and experience of early adversities may differ by sex [42,43] and family income [44,45,46], and thus, these potential confounds are controlled as well. We hypothesized that participants reporting greater early adversity would display greater increases in salivary cortisol in response to interpersonal control and relatively lower social status, compared to participants reporting lesser early adversity. Similarly, we hypothesized that participants reporting greater early adversity would experience greater increases in negative affect in response to interpersonal control and relatively lower social status, compared to participants reporting lesser early adversity.

## 2. Materials and Methods

### 2.1. Participants

Participants in this study were young adults (*n* = 160, 51.6% female) with valid salivary cortisol data. All participants in the current study were enrolled at the University of Utah, Salt Lake City, UT, United States. Participants were recruited from the undergraduate research pool (median age = 21.86 years, *SD* = 3.47, range: 17–32). The majority of participants were non-Hispanic Whites (72.7%), though a considerable proportion were Hispanic (10.3%), and Asian American (7.9%). The average education level of parents was a bachelor’s degree, and mean family income for the sample was between $75,000 and $124,999. The main effects of study manipulations without consideration of ACEs have been previously published [47].

### 2.2. Design and Procedures

The study design was a 2 (Social Evaluative Threat: High and Low) × 2 (Participant Relative Social Status: High and Low) × 2 (Partner Dominance: High and Low) × 2 (Sex: Male vs. Female) factorial design. Participants were recruited to take part in a discussion of a current event with another participant from the research pool. Unbeknownst to the participant, their discussion partner was a sex-matched confederate recording. Upon arrival to the laboratory, participants were told their interaction partner was in a nearby room and their conversation would take place through a speaker and microphone system to avoid influence from extraneous factors (e.g., appearance). Participants were also told that turn-taking was necessary because speaking could influence physiological parameters, and were told that background information would be exchanged in order to produce a more natural interaction. Participants’ relative social status was manipulated during this exchange of background information (see manipulation details below). According to published guidelines [26], participants provided three saliva samples from which cortisol was extracted (see Figure 1 and additional details below).

After being informed of study details, anonymity of data collection and storage, and the voluntary nature of participation (i.e., ability to end participation at any time), all study participants provided informed consent. Subsequently, participants completed background questionnaires assessing demographic information (e.g., race and ethnicity, family income), early adversity, and indicated their stance (e.g., either pro or con) on three topics which were publicly debated at the time of data collection. Specifically, participants indicated their stance on the effectiveness of online college courses, universal healthcare, and same-sex marriage. The topic about which participants reported having the strongest opinion was used for discussion. The confederate partner always held opposing beliefs. Following completion of baseline questionnaires, participants provided their first cortisol sample. After baseline, participants were told which topic they would be discussing and were given two minutes to prepare their thoughts. Participants then “interacted” with the confederate recording in a sequence of three, 90-s exchanges on the topic. Following the conclusion of the task, open-ended questions were used to determine if participants suspected their interaction partner was part of a manipulation. Three participants reported either suspicion or disbelief and were subsequently excluded from analyses. Finally, the purpose of the study and the use of deception were fully explained to study participants. None of the study participants reported negative reactions elicited by the study procedures. The study was conducted in accordance with the Declaration of Helsinki, and approved by the Institutional Review Board of the University of Utah (protocol 350 #00035022, approval date: 21 December 2011).

### 2.3. Manipulated Conditions

#### 2.3.1. Social Evaluative Threat

Social evaluative threat levels were manipulated in a manner similar to the TSST [26,48]. Specifically, participants randomly assigned to the high-threat condition received instructions stating that their interaction would be evaluated for skill, intelligence, competence, and accuracy of information. Additionally, a non-responsive experimenter sat facing the participant during the interaction task, supposedly making these ratings. Participants randomly assigned to the low-threat condition received instructions which specifically stated that the content of the interaction task would not be evaluated, and that participants should focus only on speaking clearly. There was no observer in the low-threat condition.

#### 2.3.2. Participant Relative Social Status Manipulation

Under the guise of facilitating a more natural conversation, participants exchanged background information with their interaction partner. The information the participant received from his or her partner was manipulated by experimenters to indicate that the partner’s ratings of subjective and objective status were either considerably higher or considerably lower than the participant’s own ratings on these same items (described in the Measures section). Specifically, participants randomly assigned to the relatively higher social status condition received background information which indicated their partner’s ratings of objective and social status were three tiers below their own ratings (or the maximum possible). Participants randomly assigned to the relatively lower social status condition received background information which indicated their partner’s ratings of objective and social status were three tiers above their own ratings (or the maximum possible). Manipulation checks confirmed the success of this manipulation [47].

#### 2.3.3. Partner Dominance Manipulation

All participants interacted with a sex-matched, pre-recorded confederate partner. The recordings for all conditions were of the same male and female confederate. Participants randomly assigned to the dominant partner condition interacted with a confederate partner who spoke in a confident, assertive, and definitive manner. Participants randomly assigned to the submissive partner condition interacted with a confederate partner who engaged hesitantly, deferentially, and lacked certainty. Manipulation checks confirmed the success of this manipulation [47].

### 2.4. Measures

#### 2.4.1. Social Status

Self-reported ratings of family income and educational attainment were used to measure objective social status. Family income was reported using sixteen possible income brackets ranging from “$0–$4999” to “greater than $500,000”. Response options for educational attainment for participants and their parents ranged from “partial high school” to “graduate degree.” [47]. Subjective social status was measured using the MacArthur Scale of Subjective Social Status (SSS); [49]. The SSS includes two visual analog ladder scales, one assessing perceived social status within the United States, and one assessing perceived social status within the individual’s community. Participants rate their social status relative to the United States by selecting one of the 10 ladder rungs, with the 10th (highest) rung representing individuals with the highest income, best education, and best careers. Participants similarly rate their social status relative to their own community by selecting one of 10 ladder rungs, with the 10th (highest) rung representing the individuals in their community with the highest social standing. The SSS has demonstrated good test–retest reliability [50] and consistent associations with objective measures of social status [51,52] in previous research.

#### 2.4.2. Adverse Childhood Experiences

ACEs were assessed using a 10-item version of the Risky Families Questionnaire [53]. Participants rated the frequency with which they experienced abuse and neglect, as well as the magnitude of household chaos and dysfunction (e.g., chaotic home, witnessed violence, verbal conflict, substance-abusing household member) between ages 5 and 15 on a 5-point scale from 1 (not at all) to 5 (very often). Higher scores indicate a greater number of ACEs. The 10-item version used in this study differs from the original 13-item version by assessing for general conflict in the home rather than between specific family member combinations (e.g., parents and siblings, participant and siblings) [54]. The 13-item version was validated against clinical interviews, and demonstrated high reliability and agreement [53]. The questionnaire has been shown to be reliably associated with adverse mental and physical health outcomes [54,55,56]. Previous work has reported good internal consistency (Cronbach’s α = 0.77–0.85) [57]. Cronbach’s alpha from our sample was also good (α = 0.78).

#### 2.4.3. Cortisol Reactivity

Cortisol levels were assessed via saliva, and salivary cortisol has been shown to be highly correlated (*r* > 0.90, *p* < 0.001) with metabolically active (free) blood cortisol levels [58,59,60,61]. Consistent with previous work [62] and timeframe recommendations for observing stress-related changes in cortisol [27,28,63], saliva samples were collected following baseline, and 25- and 35-min after beginning the interaction task (Figure 1). Sarstedt cortisol salivettes were used to collect samples before being stored using standard procedures. Immuno-assays (IBL-International) analysis was performed at the Technical University of Dresden. This method has well-documented reliability (inter- and intra-assay coefficients of variation < 5%) and sensitivity (0.2 nmol/L) [64]. Salivary cortisol responses were calculated as task-baseline change scores [65]. More specifically, three change scores for salivary cortisol were calculated: (1) the second salivary cortisol sample minus baseline, (2) the third salivary cortisol sample minus baseline, and (3) the average change in salivary cortisol from baseline across both samples. We examined the average change variable as our primary indicator of cortisol reactivity. Area under the curve increase (AUCi) was also calculated from the baseline, second, and third salivary cortisol samples [66]. AUCi was highly correlated with average salivary cortisol change (*r* = 0.99, *p* < 0.001).

#### 2.4.4. Changes in Negative Affect

To assess changes in anxiety and anger, participants completed a 12-item state anger and anxiety measure modified from the State-Trait Personality Inventory [67] shortly before and after the interaction task. Previous research has demonstrated these measures are sensitive to stressful laboratory manipulations [68], and they were reliable in this sample (αs > 0.75). We also used a 10-item measure of state self-conscious emotions originally derived from the Positive and Negative Affect Schedule [69]. The items on this measure ask participants to what extent they are feeling ashamed, embarrassed, humiliated, self-conscious, foolish, stupid, defective, awkward, exposed, and defeated [70,71]. Items were highly correlated in the authors’ original sample and in our current sample (Cronbach’s alpha = 0.85). Positive change scores reflect increases from baseline to post-task.

### 2.5. Overview of Analyses

Bivariate correlations were first conducted to examine the relationships between variables of interest, and to examine the associations between cortisol reactivity and cortisol AUC. There were two outliers on the Risky Families Questionnaire, which were winsorized to meet normality assumptions. Responses on this measure were then mean-centered to aid the interpretation of interactions. Initial analyses of variance (ANOVA) were performed to determine if ACEs significantly differed by condition (e.g., high and low status). Subsequent 2 × 2 × 2 ANOVA analyses were performed to determine if ACEs interacted with interpersonal stress to affect salivary cortisol reactivity. Given the natural daily rhythm of cortisol and previous evidence, time since waking [72], body mass index (BMI), sex [73], and family income [74] were entered as covariates. Additionally, previously reported findings from these data revealed that evaluative threat and partner dominance manipulations had an interactive effect on cortisol reactivity [47], so this interaction term was also added as a covariate in analyses examining cortisol. Similar multivariate analyses were performed to examine whether ACEs influenced changes in negative affect in response to interpersonal stressors. All analyses were conducted using SPSS version 25 [75].

## 3. Results

Descriptive statistics for variables of interest and covariates are shown in Table 1. Consistent with previous research, ACEs and family income were negatively associated in our sample [76,77]. The mean total score on the Risky Families Questionnaire was 22 (*SD* = 6.1) of the possible 50, which is higher than those found in previous large survey samples [78,79], but lower than those found in inpatients samples [80]. See Appendix A for the item content of the modified Risky Families Questionnaire used in the current study and item-level descriptives. The mean body mass index (BMI) in the sample was 23 (*SD* = 3.4), similar to mean BMI levels reported in other college-aged samples [81].

Additionally, preliminary analyses revealed one failure of randomization: participants randomized to the high social evaluative threat condition reported significantly fewer ACEs compared to participants randomized to the low social evaluative threat condition (F(3,161) = 2.825, *p* = 0.040). There were no differences in ACEs across the other two manipulations.

### 3.1. Cortisol Reactivity

Multivariate analyses controlling for time since waking, sex, and family income revealed significant main effects of relative social status and ACEs on cortisol change. Specifically, relatively lower social status and reports of greater ACEs were associated with greater cortisol reactivity, independently (Table 2; Figure 2a).

Further, the interaction of relative social status and ACEs was statistically significant. The combination of greater ACEs and lower social status resulted in the greatest cortisol response, whereas the combination of greater ACEs and higher social status resulted in the smallest cortisol response. Differences in social status did not elicit significantly different cortisol responses for participants who reported fewer ACEs (see Figure 2a). Subsequent multivariate analyses adjusting for BMI changed results only slightly (Δ$partial-\eta^{2}$ = 0.001). The interaction between ACEs and exposure to a dominant (vs. submissive) interaction partner was nonsignificant. Notably, interactions are statistically adjusted for average reactivity across all participants (the constant), as well as the main effects of the other manipulations, including reactivity to the presence (vs. absence) of social-evaluative threat [82].

### 3.2. Negative Affect Reactivity

Results of analyses examining changes in negative affect revealed a consistent main effect of ACEs. Greater ACEs were associated with significantly greater increases in anxiety (partial-$\eta^{2}\;$ = 0.34), anger (partial-$\eta^{2}\;$ = 0.31), and self-conscious emotions (partial-$\eta^{2}\;$ = 0.34) (Table 2). Additionally, the results of analyses examining changes in state anxiety revealed a significant interaction between ACEs and relative social status, as well as a significant interaction between ACEs and partner dominance. The combination of greater ACEs and relatively higher social status resulted in the greatest increase in anxiety, whereas the combination of greater ACES and relatively lower social status resulted in the smallest increase in anxiety. Differences in relative social status did not elicit significantly different changes in anxiety for participants who reported fewer ACEs (see Figure 2b). Additionally, for individuals reporting fewer ACES, interacting with a dominant partner elicited the greatest increase in anxiety, and interacting with a submissive partner elicited the smallest increase in anxiety (see Figure 3a). Differences in partner behavior did not elicit significantly different changes in anxiety for individuals reporting greater ACEs. Results of analyses examining changes in anger revealed a significant interaction between ACEs and dominant (vs. submissive) partner behavior. Though interacting with a dominant partner elicited greater anger than interacting with a submissive partner in general, this pattern was more pronounced for participants reporting greater ACEs (see Figure 3b). No significant interaction effects were found between ACEs and the manipulations with respect to self-conscious emotions.

## 4. Discussion

In the current study, we examined whether early adversity within the family (e.g., abuse, neglect, exposure to violence) was associated with changes in cortisol and negative affect in response to acute interpersonal stressors reminiscent of experiences of powerlessness, devaluation, and lack of control that characterize many early adversities. We also manipulated social-evaluative threat as a test of whether, similar to findings in the PTSD literature, these specific interpersonal stressors may be more germane to stress reactivity in the context of early adversity.

Results revealed a main effect of early adversity on cortisol reactivity, as well as changes in negative affect, such that greater early adversity was associated with greater cortisol and negative affective reactivity (anger, anxiety, self-conscious emotions) across manipulated conditions. Whereas past studies have shown mixed findings with respect to early adversity and cortisol reactivity, we may find positive effects in this particular study because two of our three fully crossed manipulated variables resemble stimuli associated with early adversity (exposure to dominance and low social status or valuation); thus, most participants received at least one of these interpersonal stressors. The finding that early adversity was consistently associated with changes in negative affect is consistent with past findings in the literature. For example, the association between early adversity and increased self-conscious emotions during the interaction task resembles results found in prior investigations linking early neglect with higher levels of shame in children [83], and interpersonal rejection and exclusion with increased self-conscious emotions and heightened negative emotionality when exposed to interpersonal stress [84].

Examination of interactions between early adversity and each of the three manipulations served as tests of whether participants’ exposure to early adversity may influence their affective or cortisol response to any one of the three manipulations, accounting for effects of the other two. We expected that one or both of the manipulations thought to resemble interpersonal characteristics of early adverse experiences (exposure to dominance and low social status or valuation) may result in larger increases in cortisol and negative affect for participants who experienced more early adversity, and we did not expect to see this difference in the social-evaluative threat manipulation. Consistent with expectations, results revealed no significant interactions between early adversity and exposure to social-evaluative threat on changes in cortisol or negative affect. However, there were a number of significant interactions between early adversity and the other two manipulations that more closely resemble interpersonal features of early adversities. Some of these interactions were consistent with expectations and some were not. For cortisol, though there was no interaction between early adversity and exposure to partner dominant (vs. submissive) behavior, there was a significant interaction between early adversity and relative social status. Participants who reported more adverse childhood experiences displayed a significant increase in salivary cortisol when they perceived themselves to be lower (vs. higher) in social status, whereas participants reporting fewer adverse childhood experiences did not show differential cortisol reactivity to social status threat. This interaction supports the hypothesis that participants with greater early adversity may be more reactive to interpersonal stressors reminiscent of early adverse experiences, and the effect size of this interaction was large according to typical conventions [85].

Early adversity also interacted with relative social status on changes in anxiety during the task, but this interaction did not mirror the interactive effects on cortisol. Instead, participants reporting more adverse childhood experiences reported a greater increase in anxiety when they perceived themselves to be higher (vs. lower) in social status. A significant interaction between early adversity and partner dominance (vs. submission) also emerged for anxiety, with only participants reporting less early adversity showing a significant difference in anxiety in response to dominant (vs. submissive) partner behavior. These two interactions do not support the hypothesis that participants with greater early adversity may be more emotionally reactive to interpersonal stressors reminiscent of early adverse experiences. Lastly, partner dominance also interacted with early adversity to predict changes in anger, but the pattern of interaction findings for anger was not consistent with the pattern of interaction findings for anxiety. Instead, participants who reported greater early adversity showed larger discrepancies in anger associated with dominant (vs. submissive) partner behavior. This interaction supports the hypothesis that participants with greater early adversity may be more reactive to interpersonal stressors reminiscent of early adverse experiences.

Taken together, results partially support the hypothesis that participants with greater early adversity may be more reactive to interpersonal stressors reminiscent of early adverse experiences. Although threats to social status evoked particularly large cortisol responses for individuals with more adverse childhood experiences, dominant (vs.) submissive partner behavior did not. However, dominant (vs. submissive) partner behavior did evoke particularly large differences in anger for individuals with more early adversity, but we did not see this same pattern for anxiety or self-conscious emotions. Experiences of early adversity may render interpersonal dominance particularly anger-inducing due to differences in how interpersonal information is processed. For example, social information processing theory [86,87] posits that harsh early experiences lead to difficulties encoding and thus appropriately responding to external cues, resulting in frustration and high levels of hostility, anger, and aggression [88]. Adverse childhood experiences have also been associated with perceptual sensitivity to anger [89], which evokes anger and self-protection in return [88], and the lack of warmth (though not hostility) expressed by dominant partners may have been perceived as hostility by participants with greater early adverse experiences.

Though results suggest that lower social status during interpersonal interactions may increase cortisol reactivity for individuals high in early adversity, they also suggest that lower social status during interpersonal interactions may attenuate anxiety reactivity for these individuals (Figure 2b). In this study, we manipulated relative social status with direct feedback about family socioeconomic status relative to an interaction partner. Insomuch as this manipulation evokes general perceptions of value relative to others, the results may also suggest that interpersonal behaviors that communicate valuing others, such as displays of respect, might attenuate physiological responses for individuals high in early adversity. Similar logic may apply to the finding that submissive partners evoked particularly low levels of anger for individuals high in early adversity (Figure 3b). In this case, results may suggest that interpersonal behaviors that communicate interest (rather than dismissal), such as listening (rather than directing), attenuate angry reactions for individuals high in early adversity.

Lastly, previous research has consistently revealed associations between adverse childhood experiences and indices of lower social status, suggesting that individuals who experience more early adversity are also more likely to experience lower social status. The results of the current study suggest that low social status elicits exaggerated cortisol responses for individuals with more adverse childhood experiences. Thus, social status threats are likely experienced with greater frequency, as well as greater severity, and may constitute an important form of chronic stress related to disease risk for individuals with more adverse childhood experiences.

### Limitations

Our measure of adverse childhood experiences is retrospectively self-reported, and there may be inaccuracies in reporting. However, previous work has emphasized the benefits of retrospective reports compared to objective methods of assessing adverse childhood experiences (e.g., reported or verified accounts of abuse) [90]. Specifically, objective methods only capture a subset of individuals whose adverse childhood experiences were also reported. When direct assessment of the early family environment is not available, retrospective reports are suitable. Further, the adverse childhood experiences examined in the current study (e.g., family verbal conflict, household chaos, and dysfunction) are exceedingly difficult to assess objectively when direct observation of family interactions is not possible.

We cannot be certain that the results of our study are generalizable to other populations. Individuals reporting a high number of ACEs (e.g., greater than one standard deviation above the study mean) in the wealthy, college-aged sample examined here may not be representative of individuals who experienced a high number of ACEs in the general population. For example, our analyses may be comparing individuals who experienced a low number of ACEs to those who experienced a moderate number of ACEs at the population level. Lastly, our results, particularly the reactivity patterns observed for those who experienced greater adverse childhood experiences, may also be influenced by selection effects in this college population. Future work should investigate differences in adverse childhood experiences between college students and community-dwelling, age-matched subjects to determine the extent to which the results reported here can be generalized.

## 5. Conclusions

The results of this study suggest that early adverse experiences are associated with greater cortisol reactivity to interpersonal stress, with perceptions of social status evoking particularly high levels of cortisol for individuals who experienced more adverse childhood experiences. Additionally, adverse childhood experiences were associated with greater increases in all three forms of negative affect (anxiety, anger, self-conscious emotions), with exposure to a dominant partner evoking particularly high levels of anger for participants who reported greater adverse childhood experiences. Although future research is needed with larger samples, these results offer preliminary evidence that participants with greater early adversity may be more reactive to interpersonal stressors reminiscent of early adverse experiences. These interpersonal exposures may represent conditioned stimuli similar to research findings in the PTSD literature, in that exposure to stressors more closely related to past trauma are more likely to show differential reactivity associated. Given the interpersonal nature of most adverse childhood experiences, conceptualizing and measuring associations with stress physiology in an interpersonal context may more closely capture the psychological and biological embedding of these early experiences.

## Figures and Tables

**Figure 1 ijerph-19-05934-f001:**
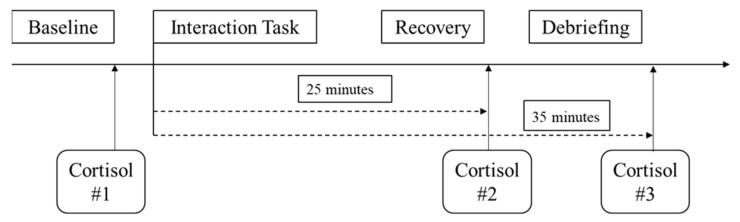
Laboratory Task Protocol and Salivary Cortisol Measurement Timing.

**Figure 2 ijerph-19-05934-f002:**
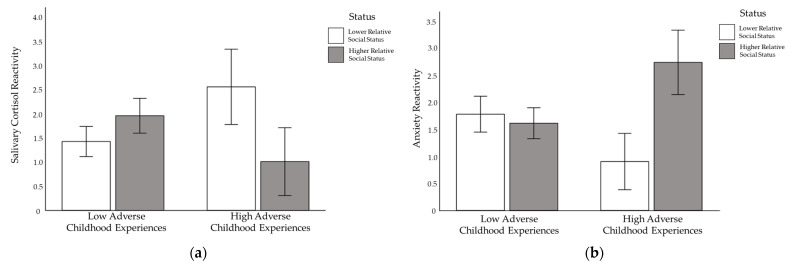
The Interaction of Early Adverse Experiences and Relative Social Status on (**a**) Salivary Cortisol Reactivity and (**b**) Anxiety Reactivity. Note. Bars represent 1 standard error.

**Figure 3 ijerph-19-05934-f003:**
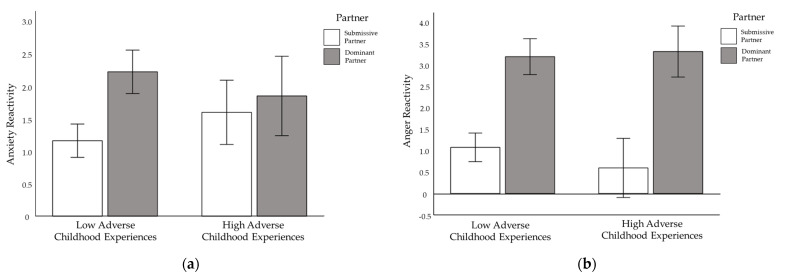
The Interaction of Early Adverse Experiences and Partner Behavior on (**a**) Anxiety Reactivity and (**b**) Anger Reactivity. Note. Bars represent 1 standard error.

**Table 1 ijerph-19-05934-t001:** Descriptive Statistics and Bivariate Correlations between Sample Characteristics and Outcomes of Interest.

	1	2	3	4	5	6	7	8
1. Age	--							
2. BMI	0.202 **	--						
3. Family Income	−0.078	0.100	--					
4. ACEs	0.045	−0.102	−0.193 *	--				
5. Cortisol Reactivity	0.028	0.074	−0.112	0.027	--			
6. Anxiety Reactivity	−0.104	−0.183 *	0.115	0.043	0.082	--		
7. SCE Reactivity	−0.021	−0.212 **	0.034	0.127	0.038	0.403 **	--	
8. Anger Reactivity	0.109	−0.025	0.049	0.078	0.045	0.450 **	0.199 *	--
*M*	21.86	23.01	6.97	21.58	1.69	1.70	1.37	2.14
*SD*	3.47	3.39	2.73	6.10	4.50	3.32	5.74	4.10

Note. Family income of 6.9 corresponds to income levels between $75,000 and $125,000. Salivary cortisol values are presented in nmol/L. ACEs = adverse childhood experiences, BMI = body mass index, SCE = self-conscious emotions. * *p* < 0.05, ** *p* < 0.01.

**Table 2 ijerph-19-05934-t002:** Results of Multivariate Analyses Examining the Associations Between Early Adverse Experiences and Study Manipulations on Changes in Salivary Cortisol and Negative Affect.

Predictor	Cortisol Reactivity			Anxiety Reactivity			SCE Reactivity			Anger Reactivity		
	*F*	*p*	η^2^	*F*	*p*	η^2^	*F*	*p*	η^2^	*F*	*p*	η^2^
Time Since Waking	0.07	0.798	0	--	--	--	--	--	--	--	--	--
BMI	0.89	0.347	0.01	--	--	--	--	--	--	--	--	--
Family Income	4.51	0.036	0.05	0.09	0.762	0	0.01	0.942	0	0.73	0.395	0.01
Sex	0.02	0.884	0	4.05	0.047	0.04	0.32	0.571	0	13.17	<0.001	0.12
Status	5.28	0.024	0.05	1.76	0.187	0.02	2.16	0.145	0.02	0	0.997	0
Dominance	0	0.946	0	0.05	0.815	0	2.19	0.142	0.02	7.63	0.007	0.08
SET	1.73	0.192	0.02	2.51	0.117	0.03	1.28	0.261	0.01	3.87	0.052	0.04
ACEs	1.75	0.028	0.32	1.98	0.010	0.34	1.94	0.012	0.34	1.69	0.037	0.31
Status∗Sex	--	--	--	--	--	--	--	--	--	0.15	0.695	0
Status∗ACEs	2.27	0.010	0.25	2.01	0.025	0.23	0.84	0.621	0.11	1.54	0.111	0.19
Dominance∗ACEs	0.96	0.504	0.13	1.95	0.027	0.24	0.89	0.574	0.12	2.21	0.011	0.26

Note. BMI = body mass index, Status = relative social status (relatively higher or lower), Dominance = partner’s behavior (dominant or submissive), SET = social evaluative threat, ACEs = adverse childhood experiences, SCE = self-conscious emotions. Previous analyses revealed that the interaction between status and sex was a significant predictor of changes in anger, so this interaction is controlled in analyses predicting anger. Participants randomized to the high social evaluative threat condition reported significantly fewer adverse childhood experiences compared to participants randomized to the low social evaluative threat condition (F(3,161) = 2.825, *p* = 0.040). Given this failure of randomization, we controlled for the main effect of the SET manipulation, but did not examine further interactions with this variable.

## Data Availability

The data presented in this study are openly available in the Harvard Dataverse at https://doi.org/10.7910/DVN/XWRMAK (accessed on 10 March 2022).

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
