# Peer review of "Early Adversity and Changes in Cortisol and Negative Affect in Response to Interpersonal Threats in the Laboratory"

_ijerph, 2022, doi:10.3390/ijerph19105934_

Round 1

Reviewer 1 Report

Dear authors, thank you for giving me the opportunity to review your manuscript: “Early adversity and cortisol reactivity to threats to social standing”. I think that this manuscript can meaningfully contribute to the literature. I suggest the following major adjustments:

Introduction

  • Line 30. Typo: “responding and deficits in emotion regulation [6-9]. which contribute to”
  • Line 32. Typo: “neurodevelopmental alterations [12-14]. in a”
  • Line 39: The authors note “a recent meta-analysis examining cortisol”, but the reference is from 2017. We cannot consider it a recent study
  • Line 48-50: Please, provide a reference.
  • The introduction would benefit from a deeper reflection and the inclusion of a revision on the relationship between variables.
  • The introduction is very poor and does not present sustainability for carrying out the study.

Significant investment by the authors is needed in the introduction.

  • Please insert a theoretical model to explain the association between the variables.

Materials and Methods

  • Please explain better how the topics of conversation were chosen and what were the topics of conversation.
  • The Social Evaluative Threat should be better explained
  • The Participant Relative Status Manipulation should be better explained
  • The Partner Dominance Manipulation should be better explained
  • “Changes in Negative Affect”: Authors should provide the name and references of the instruments.
  • All ethical considerations should be explained more.

Results

  • What kind of adverse childhood experiences did the participants report? There are several types of adverse childhood experiences, and the different forms of adverse childhood experiences imply different impacts.

Discussion

  • The authors discuss some of their study results, but the relevance of these results remains to be discussed. Please add the implications of this study for practice.
  • The discussion of associations between variables in the study is mainly limited to other studies addressing similar associations. However, a much more important issue is the lack of developed hypotheses and associated model testing against which to discuss the findings, which might add to the extant literature (e.g., Line 277: “participants who reported a greater number of adverse childhood experiences displayed a significant increase in salivary cortisol when they perceived themselves to be lower in social status, and the effect size of this interaction was large according to typical conventions [55].”).

Author Response

Reviewer #1

  • Line 30. Typo: “responding and deficits in emotion regulation [6-9]. which contribute to”
  • Line 32. Typo: “neurodevelopmental alterations [12-14]. in a”
  • Line 39: The authors note “a recent meta-analysis examining cortisol”, but the reference is from 2017. We cannot consider it a recent study
  • Line 48-50: Please, provide a reference.

Response: We have made these editorial corrections.

  • The introduction would benefit from a deeper reflection and the inclusion of a revision on the relationship between variables.
  • The introduction is very poor and does not present sustainability for carrying out the study. Significant investment by the authors is needed in the introduction.
  • Please insert a theoretical model to explain the association between the variables.

Response: We appreciate the reviewer’s direct feedback, which was in line with concerns from Review #2 as well. We have heavily edited the introduction to include a much clearer rationale for the study and more detailed information on past studies that support this rationale. We hope you find it improved and clear.

  • Please explain better how the topics of conversation were chosen and what were the topics of conversation.

Response: The description of the topics and the rationale for selecting these topics has been revised on lines 147-148 and now appears as “. . . and indicated their stance (e.g., either pro or con) on three topics which were publicly debated at the time of data collection. Specifically, participants indicated their stance on the effectiveness of online college courses, universal healthcare and same-sex marriage.”

  • The Social Evaluative Threat should be better explained

Response: The description of the social evaluative threat manipulation beginning on line 165 has been revised. Revisions include the addition of the statement “Social evaluative threat levels were manipulated in a manner similar to the TSST [29, 50]” on lines 166-167, and the description of the experimenter as “non-responsive” in the high threat condition, consistent with previous descriptions of the TSST. Additionally, the description of each study manipulations have been revised slightly to more clearly state that participants were randomly assigned to experimental conditions. For example, line 167 now appears as “Specifically, participants were randomly assigned to the high threat condition . . .”

  • The Participant Relative Status Manipulation should be better explained

Response: Additional details regarding the participant relative status manipulation and the experimental conditions were added beginning on line 175. Consistent with the revisions made to the social evaluative threat condition section, special attention was paid to clearly describing the differences between the high- and low relative social status conditions. Consistent with lines 166-167, the measures section of the manuscript now includes a description of the objective and subjective social status measures in section 2.4.1, lines 198-212.

  • The Partner Dominance Manipulation should be better explained

Response: The description of this manipulation (beginning on line 188) has been revised and now appears as: “All participants interacted with a sex-matched, pre-recorded confederate partner. The recordings for all conditions were of the same male and female confederate. Participants randomly assigned to the dominant partner condition interacted with a confederate partner who spoke in a confident, assertive and definitive manner. Participants randomly assigned to the submissive partner condition interacted with a confederate partner who engaged hesitantly, deferentially, and lacked certainty.”

  • “Changes in Negative Affect”: Authors should provide the name and references of the instruments.

Response: This section (beginning on line 253) has been revised to more clearly describe the 12-item anger and anxiety measure used here as a modified version of the State-Trait Personality Inventory.  Similarly, we have revised the description of our measure of self-conscious emotions, which now appears as “We also used a 10-item measure of state self-conscious emotions, which asks participants to what extent they are feeling negatively about themselves (e.g., “foolish”) at the moment [71].”

  • All ethical considerations should be explained more.

Response: The design and procedures section has been expanded to more clearly discuss the study procedures implemented to ensure ethical collection of the study data. For example, we have added the following sentence beginning on line 250: “anonymity of data collection and storage, and the voluntary nature of participation (i.e., ability to end participation at any time), all study participants provided informed consent.”

Additionally, we have added the following information beginning on line 159: “Finally, the purpose of the study and the use of deception were fully explained to study participants. None of the study participants reported negative reactions elicited by the study procedures. The study was conducted in accordance with the Declaration of Helsinki and approved by the Institutional Review Board of the University of Utah (protocol 350 #00035022, approval date: 12/21/2011).”

  • What kind of adverse childhood experiences did the participants report? There are several types of adverse childhood experiences, and the different forms of adverse childhood experiences imply different impacts.

Response: The item content and item-level descriptives have been added as a supplementary table in the manuscript and are included on the final page of this letter as well. The following text has been added on lines 277-278: “See Supplementary Table S1 for the item content of the modified Risky Families Questionnaire used in the current study and item-level descriptives.”

  • The authors discuss some of their study results, but the relevance of these results remains to be discussed. Please add the implications of this study for practice.

Response: The discussion section has been heavily revised in line with comments from both reviewers, including implications. We do not specifically discuss implications for practice given the nature of our study, which has high internal validity but may not generalize to interactions in the real world or in specific settings. Nonetheless, we do discuss general implications of the findings, including implications if these effects generalize to everyday life.

  • The discussion of associations between variables in the study is mainly limited to other studies addressing similar associations. However, a much more important issue is the lack of developed hypotheses and associated model testing against which to discuss the findings, which might add to the extant literature (e.g., Line 277: “participants who reported a greater number of adverse childhood experiences displayed a significant increase in salivary cortisol when they perceived themselves to be lower in social status, and the effect size of this interaction was large according to typical conventions [55].”).

Response: The discussion section has been heavily edited and enriched. Results are now clearly interpreted in the context of specific hypotheses linked with specific statistical tests. We hope you find it improved and clear.

Supplementary Table S1

Item content and descriptive statistics of the modified Risky Families Questionnaire used in the current study.

Item Content

M

SD

1. How often did a parent or other adult in the household make you feel that you were loved, supported, and cared for?*

4.40

0.91

2.  How often did a parent or other adult in the household swear at you, insult you, put you down, or act in a way that made you feel threatened?

2.12

1.16

3. How often did a parent or other adult in the household express physical affection for you, such as hugging, or other physical gestures of warmth and affection?*

3.98

1.07

4. How often did a parent or other adult in the household push, grab, shove, or slap you?

1.48

0.84

5. In your childhood, did you live with anyone who was a problem drinker or alcoholic, or who used street drugs?

1.54

1.16

6. Would you say that the household you grew up in was well-organized and well-managed?*

3.95

1.09

7.  How often would you say that a parent or other adult in the household behaved violently toward a family member or visitor in your home?

1.50

0.96

8. How often would you say there was quarreling, arguing, or shouting in your home?

2.76

1.09

9. Would you say the household you grew up in was chaotic and disorganized?

1.89

1.06

10. How often would you say you were neglected while you were growing up, that is, left on your own to fend for yourself?

1.81

1.15

Note. All items rated on a 5-point scale ranging from 1 (not at all) to 5 (very often). *Reverse Scored Items

Reviewer 2 Report

This manuscript reports a study that analyses the associations between the frequency of adverse childhood experiences (ACE) and affective and cortisol reactivity under three controlled conditions: presence vs. absence of a social evaluative threat, interaction partner using a controlling vs. submissive interactional style, and interaction partner positioned in a lower vs. higher social rank. Thus, the study design was a 2X2X2 factorial design.

The study is well designed and has the potential to introduce some novelty to the existing knowledge in the field of adverse childhood experiences and its association with interpersonal stressors. However, I have a main concern regarding the conceptual support of the study that should be considered before the manuscript is ready for publication.

From my point of view, the introduction section (and consequently the discussion section) lack conceptual and/or empirical support and justification. Some specific aspects illustrating my point of view, and especially important to justify the methodological options regarding the selection of the three manipulated variables, are as follows:

  • In the Introduction section, no conceptual support is presented to include affective reactivity (anger, anxiety, and self-conscious emotions) as an outcome variable. Interestingly, affective reactivity is also omitted from the title despite being a variable of interest.
  • The expectations presented in the last paragraph of the introduction section regarding the role of social rank on ACE are not well supported, either conceptually or empirically.
  • Also, the rationale underlying the use of presence vs. absence of a social evaluative threat as a covariate is not justified enough; in fact, it is not clear if it is considered a control variable or an experimental condition. In the abstract, the authors wrote “… We further examine whether such effects are significant over and above effects of a social evaluative threat manipulation …” (covariate?), while at the end of the introduction authors wrote “… over and above commonly used laboratory stressors, we also manipulate social-evaluative threat,” (experimental condition?).

Additionally, there are some minor remarks that the authors should consider to improve the manuscript in terms of clarity. Please see my specific comments below.

  • Regarding the formulation of the study objective as “whether ACE potentiate cortisol reactivity to stressors…”, I find “potentiate” to be inaccurate.
  • On page 2, the study design is described as a 2X2X2X2 factorial design. However, on page 5 and throughout the manuscript, participants’ sex is never studied as an experimental condition. Please clarify.
  • The covariates included in the study are presented only on page 5 without any previous justification, which should have been done in the introduction section.
  • Participants are young adults. However, the upper value of the age range is 54. – is this right? If so, is this an outlier? In this case, perhaps the authors should consider deleting this subject.
  • Dos the reference number 25 apply to the social evaluative threat experimental condition/procedure?
  • The names of the variables should be presented consistently. For instance, is “relative status manipulation” a synonym for “perceive social rank” and “perceptions of social status”? Figure 2 and Table 2 present the same variables with different names.
  • The Discussion section should be revised according to the modifications included in the introduction.

I hope the authors find my comments and suggestions helpful for improving this manuscript.

Author Response

Reviewer #2

The study is well designed and has the potential to introduce some novelty to the existing knowledge in the field of adverse childhood experiences and its association with interpersonal stressors. However, I have a main concern regarding the conceptual support of the study that should be considered before the manuscript is ready for publication.

From my point of view, the introduction section (and consequently the discussion section) lack conceptual and/or empirical support and justification. Some specific aspects illustrating my point of view, and especially important to justify the methodological options regarding the selection of the three manipulated variables, are as follows:

  • In the Introduction section, no conceptual support is presented to include affective reactivity (anger, anxiety, and self-conscious emotions) as an outcome variable. Interestingly, affective reactivity is also omitted from the title despite being a variable of interest.
  • The expectations presented in the last paragraph of the introduction section regarding the role of social rank on ACE are not well supported, either conceptually or empirically.
  • Also, the rationale underlying the use of presence vs. absence of a social evaluative threat as a covariate is not justified enough; in fact, it is not clear if it is considered a control variable or an experimental condition. In the abstract, the authors wrote “… We further examine whether such effects are significant over and above effects of a social evaluative threat manipulation …” (covariate?), while at the end of the introduction authors wrote “… over and above commonly used laboratory stressors, we also manipulate social-evaluative threat,” (experimental condition?).

Response: Thank you for this clear feedback. We now provide additional justification for examining changes in negative affect as an outcome and have also edited the title as follows: Early adversity and changes in cortisol and negative affect in response to interpersonal threats.

We now also offer a more clear and cohesive logic for the rationale behind the study, which is whether early adversity is associated with changes in cortisol and negative affect in response to acute interpersonal stressors reminiscent of experiences of powerlessness and lack of control that characterize many early adverse experiences. We also manipulated social-evaluative threat as a test of whether, similar to findings in the PTSD literature, exposure to interpersonal stimuli specifically associated with early adversity may be more likely to reveal differences in stress reactivity associated with early adversity.

  • Regarding the formulation of the study objective as “whether ACE potentiate cortisol reactivity to stressors…”, I find “potentiate” to be inaccurate.

Response: The language used in this section of the manuscript (lines 100-102) has been revised and now appears as “The current study examines whether early adversity within the family (e.g., abuse, neglect, exposure to violence) is associated with changes in cortisol and negative affect in response to interpersonal stress in a sample of 165 young adults (Median age = 21).”

  • On page 2, the study design is described as a 2X2X2X2 factorial design. However, on page 5 and throughout the manuscript, participants’ sex is never studied as an experimental condition. Please clarify.

Response: Yes, participant sex is not examined in the same manner as partner dominance, relative social status, and social evaluative threat. However, we describe the design as a 2x2x2x2 design because the study was conducted to ensure an equal number of male and female participants (and sex-matched confederate partners) across experimental conditions. We have revised Table 2 beginning on line 298 to display participant sex similarly to the presentation of the study manipulations.

  • The covariates included in the study are presented only on page 5 without any previous justification, which should have been done in the introduction section.

Response: Thank you for pointing this out. We have added the following text to the end of the introduction where we describe the current study: “Because cortisol reactivity can be influenced by time of day and participants’ body mass, we control for these factors in our analyses. Further, both cortisol reactivity and experience of early adversities may differ by sex [45-46] and family income [47-48], thus these potential confounds are controlled as well.”

  • Participants are young adults. However, the upper value of the age range is 54. – is this right? If so, is this an outlier? In this case, perhaps the authors should consider deleting this subject.

Response: Yes, 54 is the upper limit of the age range, and is a statistical outlier. We agree with your suggestion of removing these outliers from the data. In total, we removed 5 participants whose age is significantly higher than the sample mean (>32 years). After removing these individuals, the results of the current study were changed slightly. Most notably, multivariate analyses revealed significant main effects of relative social status and ACEs on cortisol reactivity. These effects were previously marginally statistically significant (ps<.09). The pattern of significant interactions was not changed by the removal of age outliers.

Additional changes related to the removal of these 5 participants include:

- The sample size and sample characteristics beginning on line 117 have been revised.

- The results of multivariate analyses examining cortisol reactivity have been revised to discuss the significant main effects beginning on line 294.

- Table 1 (beginning on line 282) and Table 2 (beginning on line 288) have both been revised to depict the results of analyses after the removal of outliers.

- The discussion of analyses examining changes in negative affect (beginning on line 320) have been revised.

- Figure 2 (line 340) and Figure 3 (line 347) have been revised.

  • Does the reference number 25 apply to the social evaluative threat experimental condition/procedure?

Response: Reference number 25 (now reference number 50 is provided in relation to the relationship characteristics linked to ACES. However, the discussion of the social evaluative threat manipulation was expanded considerably to address Reviewer #1’s feedback. These revisions include more detailed discussion of the social evaluative threat manipulation and reference [50] describing the social evaluative threat manipulation characteristics in the Trier Social Stress Test. We hope these changes address any concerns related to reference number 50.

  • The names of the variables should be presented consistently. For instance, is “relative status manipulation” a synonym for “perceive social rank” and “perceptions of social status”? Figure 2 and Table 2 present the same variables with different names.

Response: The language used to refer to the relative social status manipulation has been revised for consistency. Additionally, Figure 2 and Table 2 have been revised to present the manipulated conditions and outcomes of interest using consistent language. Table 1 has also been revised with this in mind.

  • The Discussion section should be revised according to the modifications included in the introduction.

Response: The discussion section has been heavily edited and enriched in line with changes to the introduction. Results are now clearly interpreted in the context of specific hypotheses for which rationale is provided in the introduction.

Round 2

Reviewer 1 Report

The authors responded to the recommendations, improving the document. The article makes a good contribution to scientific knowledge in this field.

Author Response

Reviewer #1

The authors responded to the recommendations, improving the document. The article makes a good contribution to scientific knowledge in this field

Response: We appreciate the time spent reviewing our submission and the helpful feedback provided during the previous round of revisions.  

Reviewer 2 Report

I highly appreciated the authors’ effort to improve the manuscript, taking into consideration the suggestions and recommendations made by the reviewers.

I still think that this study has the potential to introduce some novelty to the existing knowledge in the field of adverse childhood experiences and its association with interpersonal stressors. The manuscript is now much clearer, and the conceptual background is more robust. However, from my point of view, some issues should still be brought into consideration

  1. The reader is only faced with the study expectations and hypotheses in the discussion section. In fact, the formulation of hypotheses would very much contribute to clarifying the study’s objectives and design and should be included at the end of the Introduction section.
  2. The Introduction section presents a minimalistic rationale for the use of negative affect (anger, anxiety, and self-conscious emotions) as an outcome variable in this study. As far as I can understand, the rationale for the consideration of affective responses to stressors is presented only in the first paragraph of the manuscript, despite new references have been included in the discussion section (88 and 89). Please note that, on the contrary, the rationale for the other study outcome (cortisol reactivity) is presented in a greater depth.
  3. Self-conscious emotions (SCE) consist of emotional experiences requiring objective self-awareness and the ability to evaluate behavior according to internalized standards, rules, and goals (Lewis, 1992), for instance, guilt, shame, or pride. In this study, the authors do not describe the assessment of SCE. Readers are informed that the participants completed a measure of negative feelings about themselves, which is a somewhat limited operationalization for such a complex construct.
  4. The title could be improved by referring the reader to the experimental nature of the design. I would suggest something like “the effects of early family adversity on ...”

Minor remarks:

Please check the number of participants stated in the introduction section (165) and the method section (160). In addition, it is not necessary to include the mean age of participants in the introduction section.

Author Response

Reviewer #2

I highly appreciated the authors’ effort to improve the manuscript, taking into consideration the suggestions and recommendations made by the reviewers.

I still think that this study has the potential to introduce some novelty to the existing knowledge in the field of adverse childhood experiences and its association with interpersonal stressors. The manuscript is now much clearer, and the conceptual background is more robust. However, from my point of view, some issues should still be brought into consideration

  1. The reader is only faced with the study expectations and hypotheses in the discussion section. In fact, the formulation of hypotheses would very much contribute to clarifying the study’s objectives and design and should be included at the end of the Introduction section.

Response: The study hypotheses have been clearly stated on lines 120-125.

  1. The Introduction section presents a minimalistic rationale for the use of negative affect (anger, anxiety, and self-conscious emotions) as an outcome variable in this study. As far as I can understand, the rationale for the consideration of affective responses to stressors is presented only in the first paragraph of the manuscript, despite new references have been included in the discussion section (88 and 89). Please note that, on the contrary, the rationale for the other study outcome (cortisol reactivity) is presented in a greater depth.

Response: The introduction has been expanded to provide additional information regarding the associations between early adversity and negative affect, as well as the direct and indirect effects negative affect may have on physical health.

  1. Self-conscious emotions (SCE) consist of emotional experiences requiring objective self-awareness and the ability to evaluate behavior according to internalized standards, rules, and goals (Lewis, 1992), for instance, guilt, shame, or pride. In this study, the authors do not describe the assessment of SCE. Readers are informed that the participants completed a measure of negative feelings about themselves, which is a somewhat limited operationalization for such a complex construct.

Response: The description of this measure has been expanded to describe the creation of this measure as well as the specific items asked. Although self-conscious emotions differ from one another and are complex, this measure assess the construct quite directly. Beginning on line 261, this section now appears as:

“We also used a 10-item measure of state self-conscious emotions originally derived from the Positive and Negative Affect Schedule [67-72]. The items on this measure ask participants to what extent they are feeling ashamed, embarrassed, humiliated, self-conscious, foolish, stupid, defective, awkward, exposed and defeated [73].”

  1. The title could be improved by referring the reader to the experimental nature of the design. I would suggest something like “the effects of early family adversity on ...”

Response: The title has been revised and now appears as “Early adversity and changes in cortisol and negative affect in response to interpersonal threats in the laboratory.” We opted for this change to emphasize the experimental design of the study and the interpersonal stressor manipulations used here without describing early adversity in a manner that may be misconstrued as a study manipulation rather than a trait variable.

Minor remarks:

Please check the number of participants stated in the introduction section (165) and the method section (160). In addition, it is not necessary to include the mean age of participants in the introduction section.

Thank you for orienting us to this oversite. The sample size is now described consistently on lines 107 and 128.
